# Zeb1 promotes corneal neovascularization by regulation of vascular endothelial cell proliferation

Lei Jin[1,2,6], Yingnan Zhang[1,3,6], Wei Liang[1,2,6], Xiaoqin Lu[1], Niloofar Piri[1], Wei Wang[1], Henry J. Kaplan[1], Douglas C. Dean[1,4,5 ✉], Lijun Zhang[2 ✉] & Yongqing Liu [1,4,5 ✉]

Angiogenesis is required for tissue repair; but abnormal angiogenesis or neovascularization (NV) causes diseases in the eye. The avascular status in the cornea is a prerequisite for corneal clarity and thought to be maintained by the equilibrium between proangiogenic and antiangiogenic factors that controls proliferation and migration of vascular endothelial cells (ECs) sprouting from the pericorneal plexus. VEGF is the most important intrinsic factor for angiogenesis; anti-VEGF therapies are available for treating ocular NV. However, the effectiveness of the therapies is limited because of VEGF-independent mechanism(s). We show that Zeb1 is an important factor promoting vascular EC proliferation and corneal NV; and a couple of small molecule inhibitors can evict Ctbp from the Zeb1–Ctbp complex, thereby reducing EC Zeb1 expression, proliferation, and corneal NV. We conclude that Zeb1-regulation of angiogenesis is independent of Vegf and that the ZEB1–CtBP inhibitors can be of potential therapeutic significance in treating corneal NV.

[1] Department of Ophthalmology and Visual Sciences, University of Louisville School of Medicine, Louisville, KY 40202, USA. [2] Department of Ophthalmology, The Third People's Hospital of Dalian, Dalian Medical University, Dalian 116033, China. [3] Beijing Tongren Eye Center, Beijing Tongren Hospital, Capital Medical University, Beijing Ophthalmology & Visual Science Key Lab, Beijing 100730, China. [4] Birth Defects Center, University of Louisville School of Dentistry, Louisville, KY 40202, USA. [5] James Brown Cancer Center, University of Louisville School of Medicine, Louisville, KY 40202, USA. [6]These authors contributed equally: Lei Jin, Yingnan Zhang, Wei Liang. ✉email: dcdean01@louisville.edu; lijunzhangw@gmail.com; y0liu016@louisville.edu

Angiogenesis is an important process for tissue repair upon traumatic injuries and ischemic damages[1,2]. However, abnormal angiogenesis, also known as neovascularization (NV), causes diseases in the eye[3,4]. The cornea is an avascular tissue, an ideal model tissue to study NV because of its transparent and accessible nature[5]. Corneal NV is a common ocular NV disease that can be resulted from allograft transplant, contact wear, traumatic injury, and pathogenic infection-induced inflammation, and is a top third cause for blindness worldwide[3]. Many cytokines and growth factors secreted by local stromal and immune cells in the affected cornea promote new vessel projection from the pericorneal plexus[3]. Consistent inflammation induces corneal NV, irreversibly damages the endothelium, results in scars in the stroma, leading to decreased vision and even blindness[3]. Normal corneal transparency is based on its avascular physiology that is maintained by a large quantity of antiangiogenic factors in the cornea[3]. The mechanism underlying the inflammation-induced NV is thought to be imbalance between pro- and anti-angiogenesis factors, among which vascular endothelial growth factor (VEGF) and its receptor VEGFR are the most important intrinsic factors for initiation and progression of NV[3,6]. The molecular signal for VEGF-induction of NV is initiated when VEGF binds to the receptor tyrosine kinase (RTK) VEGFR, the auto-phosphorylation of VEGFR induced by the ligand VEGF leads to series of cascades of phosphorylation of downstream kinases to the final MAP kinase ERK and results in vascular endothelial cell (EC) proliferation to generate new vessels by sprouting[6,7].

In the injured cornea, large quantities of apoptotic cytokines and proteinases are released from the epithelium basal membrane and stromal keratocytes, leading to keratocyte death and intrusion of leukocytes[8]. The intruded immune cells secrete a large number of angiogenic cytokines including VEGF[8]. Modulation of VEGF activity could inhibit NV induction and progression[6,7]. Anti-VEGF therapy is popular for treating neovascular age-related macular degeneration (nAMD), diabetic macular edema (DME), neovascular glaucoma, and to a less extend for treating corneal NV[9–11]. The major issues of the anti-VEGF therapy include that not all patients respond to the therapy and the therapy becomes less or even not effective over time[10], suggesting that at least two mechanisms exist: VEGF-dependent and VEGF-independent. Therefore, comprehensively understanding the molecular mechanism underlying NV including tumor-related NV is critical for formulating a novel strategy to treat NV.

ZEB1 is an important transcription factor (TF) for epithelial to mesenchymal transition (EMT)[12]. Our group and other researchers find that ZEB1 is an oncogenic factor in promoting tumor cell proliferation, migration, and invasion[13–27]. Recently, ZEB1 has been associated with NV in breast cancer[28]. Depending on the interactive partner, ZEB1 can repress or promote expression of target genes[12]. In general, ZEB1 stays on associated genes to repress their expression by accommodating a histone deacetylase (HDAC) via the partner called C-terminal binding protein (CtBP) to make double-strand DNA (dsDNA) not accessible for the RNA polymerase to synthesize new mRNA of target genes[12]. If CtBP detaches from ZEB1 however, an acetyltransferase like P300 and PCAF would possibly interact with ZEB1 to form a new partnership to make dsDNA more accessible for mRNA synthesis of target genes, thereby promote their expression[12]. The maintenance of ZEB1 repression capacity depends on the ZEB1–CtBP complex integrity, or vice versa. Cofactors are required or excluded to maintain the ZEB1–CtBP complex[29]. For example, the extracellular signal-regulated kinase (ERK) can phosphorylate MAPK-regulated corepressor-interacting protein 1 (MCRIP1) to keep ZEB1–CtBP repression of gene expression[30]. Therefore, inhibition of these kinase activities by

specific small molecule inhibitors like U0126 would increase expression of target genes[30]. ZEB1 target genes include the epithelial cell-specific gene E-cadherin (*CDH1*) and cyclin-dependent kinase inhibitors (*CDKIs*), whose expression would result in inhibition of cell proliferation[12]. Here for the first time, we show that homozygous knockout of Zeb1 (Zeb1$^{-/-}$) retards vasculogenesis in embryonic mouse lungs, and partial deletion of Zeb1 reduces alkali-induced mouse corneal NV. We also show that deletion or knockdown of Zeb1 reduces mouse embryonic fibroblast (MEF) and mouse retinal microvascular endothelial cell (mRMVEC) proliferation, respectively, with no downregulation of their *Vegf* expression. As is the case with ZEB1, few small molecule inhibitors of transcription factors are known[31]. As an alternative to direct inhibition, we have taken advantage of the ZEB1 interaction with CtBP, which can be targeted[29]. We provide evidence that the ZEB1–CtBP inhibitors MTOB and NSC95397 can physically evict Ctbp from the Zeb1–Ctbp complex thereby upregulate expression of the miR-200 family, leading to reduction of Zeb1 expression, mRMVEC proliferation, and mouse corneal NV severity. We conclude that ZEB1-regulation of corneal NV is independent of VEGF and the ZEB1–CtBP inhibitors can be of potential therapeutic significance in treating ocular NV[3], and likely cancers as well.

## Results

**Zeb1-regulation of vasculogenesis in fetal mouse lungs.** Zeb1 is one of essential transcription factors in development, complete loss of Zeb1 function results in death of Zeb1$^{-/-}$ mouse embryos[32,33]. To see if Zeb1 is required for normal vasculogenesis in development, we compared the hematoxylin and eosin (H&E) paraffin sections of embryonic day 19.5 (E19.5) homozygous, heterozygous Zeb1-knockout embryos and their wild-type siblings (Zeb1$^{-/-}$, Zeb1$^{-/+}$, and Zeb1$^{+/+}$, respectively) (Fig. 1a–c). We found that the blood capillaries in Zeb1$^{-/-}$ and Zeb1$^{-/+}$ lung tissue were significantly underdeveloped compared to Zeb1$^{+/+}$ and the lung of Zeb1$^{-/-}$ was full of mesenchymal cells compared

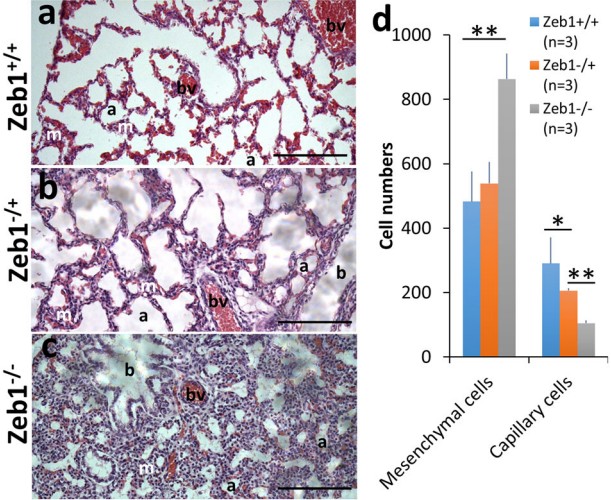

**Fig. 1 Zeb1-regulation of mouse embryonic lung development.** Representative H&E-stained paraffin lung sections of (**a**) wild-type embryos (Zeb1$^{+/+}$) at E19.5 and their (**b**) heterozygous (Zeb1$^{-/+}$) and (**c**) Zeb1 homozygous (Zeb1$^{-/-}$) knockout siblings, showing (**d**) more mesenchymal cells with a blue nucleus (m) and less capillary cells in Zeb1$^{-/-}$ knockout lungs. Capillary cells are defined as the separated red areas that may contain a single or group of red blood cells and may or may not surrounded by the mesenchymal cells. "a" denotes alveoli; "b" denotes bronchus; "bv" denotes blood vessel; "m" denotes mesenchymal cell; *$p \leq 0.05$; **$p \leq 0.01$. Scale bars represent 100 μm.

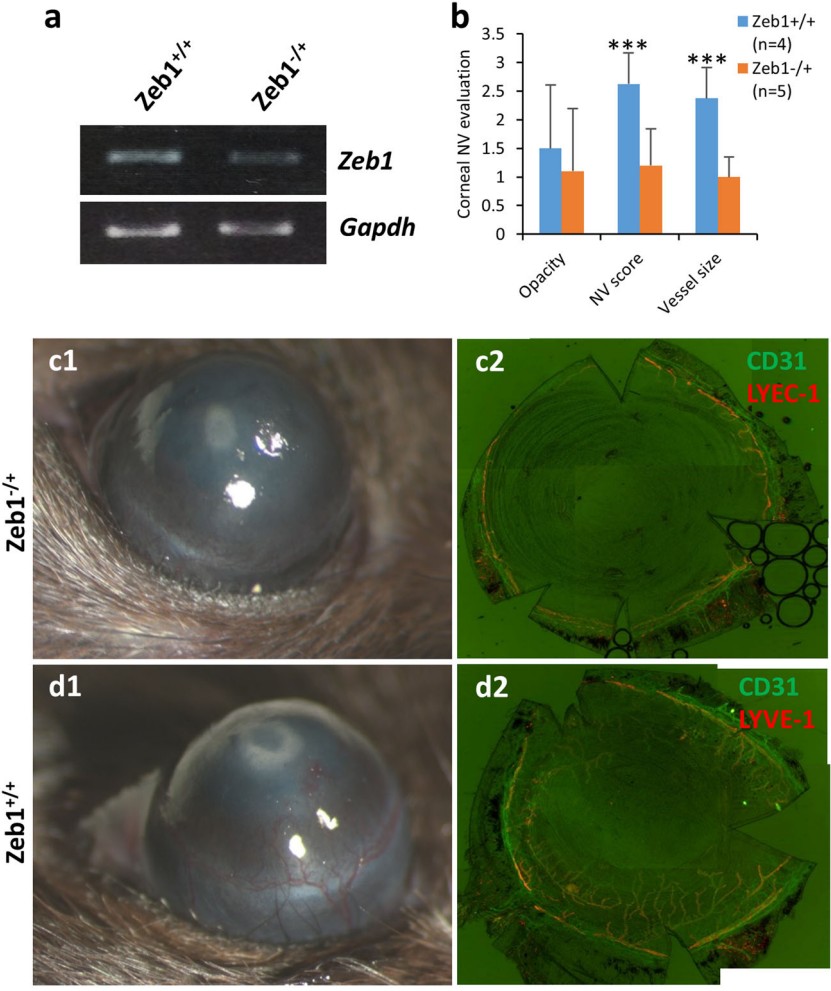

**Fig. 2 Zeb1-regulation of alkali-induced corneal NV in mice. a** Reduction of *Zeb1* mRNA in the heterozygous Zeb$^{-/+}$ corneas compared to the homozygous Zeb1$^{+/+}$ corneas. **b** Significant NV score and vessel size reduction in the Zeb1$^{-/+}$ corneas as compared to their Zeb1$^{+/+}$ siblings. **c1** A representative stereoscopic image of the Zeb1$^{-/+}$ and **d1** Zeb1$^{+/+}$ corneas and **c2–d2** their whole flat-mount corneas immunostained with the endothelium marker CD31 and the lymphatic vessel marker LYVE-1. *$p \leq 0.05$; **$p \leq 0.01$.

to Zeb1$^{+/+}$ and Zeb1$^{-/+}$ (Fig. 1a–d). This is consistent with the observation that ZEB1 was associated with NV in breast cancer[28], and it demonstrates that the attenuation of Zeb1 expression reduces blood vessel formation in the lung, and the elimination of Zeb1 is likely the cause of death of Zeb1$^{-/-}$ embryos[32].

**Zeb1 deletion reduces angiogenesis in corneal NV model mice.** To test whether Zeb1 regulates angiogenesis in adult animals, we sought to evaluate the alkali-induced corneal NV in both Zeb1$^{-/+}$ and Zeb1$^{+/+}$ mice as no Zeb1$^{-/-}$ mouse embryo would survive right before birth. We found that the partial deletion of Zeb1 significantly reduced *Zeb1* expression in the cornea detected by a real-time quantitative PCR (qPCR) (Fig. 2a) and the alkali-induced corneal angiogenesis and lymphogenesis in Zeb1$^{-/+}$ mice were significantly less severe than that in Zeb1$^{+/+}$ mice (Fig. 2b–d), suggesting that Zeb1 promotes angiogenesis in an adult tissue. Angiogenesis is dependent on vascular EC proliferation and migration[34]. To see whether Zeb1 expresses in ECs and whether the corneal NV correlates with an increased expression of Zeb1 in ECs, we compared newly formed vessels in the central corneal stroma to that in the limbus of both the alkali-burned and PBS-treated control corneas. We found that the vascular ECs of the neovascularized vessels had a higher expression of Zeb1 than that in the limbus whereas little Zeb1 was detected by

immunostaining in the vascular ECs of the PBS-treated limbus (Fig. 3a–d) and the alkali treatment increased the number of Zeb1$^+$ vascular ECs (Fig. 3c) and caused corneal NV (Fig. 3d), suggesting that new vessel formation likely needs more Zeb1 for vascular EC proliferation.

**Zeb1-regulation of NV is not through regulation of *Vegf* genes.** VEGF-regulation of angiogenesis is a well-known mechanism to initiate and promote blood vessel formation[7,34–37]. The VEGF family consists of four sister factors VEGFA/B/C/D. VEGFA/B mainly bind to two major VEGF receptors VEGFR1/2 to initiate signaling cascade in stimulating vascular EC migration and proliferation while VEGFC/D mainly bind to VEGFR3 to stimulate lymphatic vessel formation. VEGF can be an autocrine factor, but mostly a paracrine factor that bystander stromal cells secrete to affect vascular EC proliferation, migration, and invasion[7,34–37]. To determine whether Zeb1-regulation of NV is through regulation of Vegf or not, we performed a qPCR for *Vegfa, Vegfb, Vegfc, Vegfr1, Vegfr2,* and *Vegfr3* genes in Zeb1$^{+/+}$, Zeb1$^{+/-}$, and Zeb1$^{-/-}$ MEFs as corneal keratocytes are specialized fibroblast cells in the stroma. We found that knockout of Zeb1 diminished *Zeb1* expression (Fig. 4a)[18,38,39], but did not downregulate the expression of *Vegf* genes in mouse embryonic fibroblasts (MEFs) (Fig. 4b). Instead, it upregulated *Vegfa/b* and *Vegfr2* (Fig. 4b).

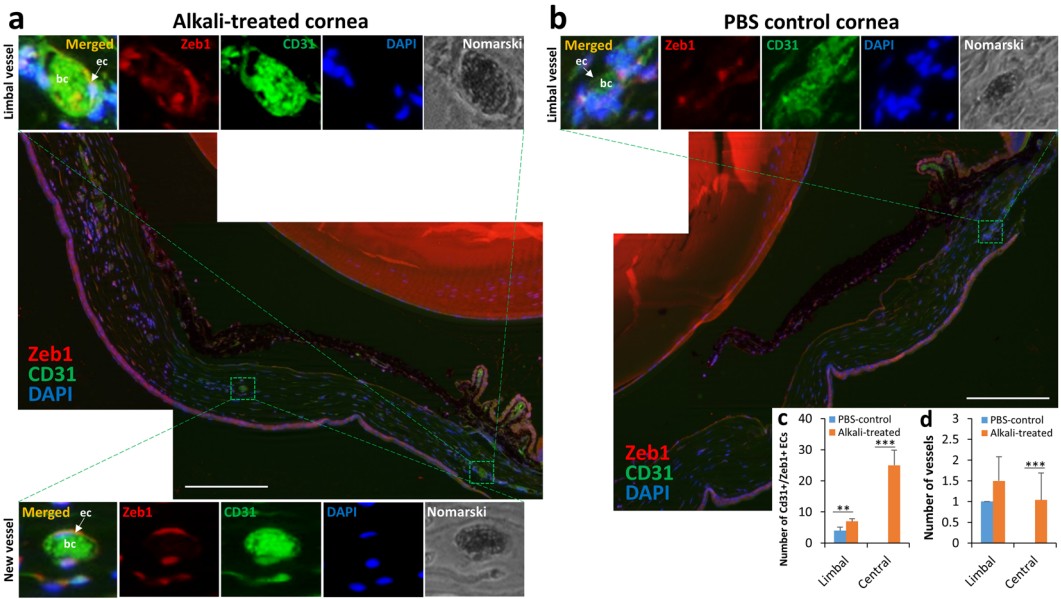

**Fig. 3 Zeb1 was expressed higher in the vascular endothelial cells (ECs) of newly formed blood vessels in the alkali-burned mouse corneas. a** A representative image of the alkali-burned cornea immunostained with Zeb1 and the EC marker CD31. Note the difference in number of blood vessels and expression of Zeb1 in the vascular ECs between the limbus and center regions of the cornea. **b** A representative image of the PBS-mocked control cornea. Note that no new vessel was observed in the center region of the cornea. **c** Number of Cd31+/Zeb1+ vascular endothelial cells and (**d**) number of vessels counted based on four cryosections of two PBS control and two alkali-treated corneas. bc blood cells, ec epithelial cells. Scale bars represent 200 μm.

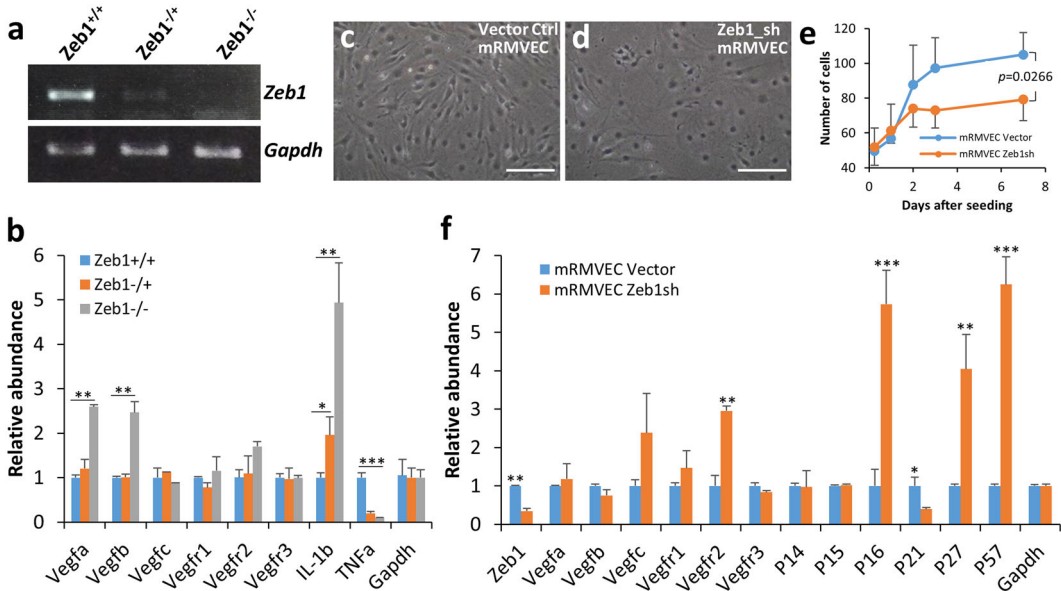

**Fig. 4 Knockout or knockdown of Zeb1 causes cellular senescence and does not downregulate *Vegf* genes in MEFs and mRMVECs. a** mRNA expression of *Zeb1* in Zeb1+/+, Zeb1−/+, and Zeb1−/− MEFs. **b** *Vegf* gene expression detected by real-time quantitative PCR (qPCR) in Zeb1+/+, Zeb1−/+, and Zeb1−/− MEFs. **c** Normal appearance of mRMVECs. **d** Senescent morphology of mRMEVCs with knockdown of Zeb1 by shRNA (Zeb1sh). **e** Knockdown of Zeb1 reduced mRMVEC proliferation rate. **f** Expression of genes detected by qPCR in the Zeb1sh mRMVECs compared to the vector control. *p ≤ 0.05; **p ≤ 0.01; ***p ≤ 0.001. Scale bars represent 100 μm.

These results suggest that Zeb1-regulation of corneal NV is independent of Vegf pathway.

**Knockdown of Zeb1 induces senescence of mRMVECs.** Loss of Zeb1 lead to quick cellular senescence of MEFs in vitro, and caused depletion of both mesenchymal and brain neural progenitor cells in vivo[40]. This early cellular senescence was linked to the de-repression of Cdk inhibitors *P15* and *P21* by Zeb1

knockout[40]. In addition, knockdown of Zeb1 by short-hairpin RNA (shRNA) significantly reduced malignant cell proliferation rates in many cancers[22,23,26,27,41], confirming that Zeb1 is a cell proliferation driver. To see whether knockdown of Zeb1 would also cause cellular senescence of vascular ECs, we transduced mouse retina microvascular endothelial cells (mRMVECs) with Zeb1 shRNA lentivirus in culture, and isolated total RNA from both vector control and Zeb1 shRNA cells for qPCR analysis. As expected, the Zeb1 shRNA-transduced cells exhibited a senescent

morphology: bigger and flat with a reduced proliferation rate compared to the vector control cells (Fig. 4c–e). The qPCR analysis showed that the Zeb1 shRNA transduction reduced *Zeb1* expression by >60% compared to the vector control; and no significant change was detected in the expression of *Vegf* genes except for an upregulation of *Vegfr2* (Fig. 4f). Knockdown of Zeb1 also significantly increased the expression of the Cdk inhibitors *P16, P27,* and *P57* (Fig. 4f). These results suggest that the cease of mRMVEC proliferation is likely caused by the induction of the Zeb1-repressed Cdk inhibitors.

**ZEB1–CtBP inhibitors functionally inactivates Zeb1 in mRMVECs.** No direct Zeb1 inhibitor has ever been identified while some indirect small molecules were suggested to inactivate ZEB1 by inhibition of ZEB1 and CtBP interaction such as MTOB and NSC95397[29,42]. MTOB, a known substrate inhibitor for CtBP, can functionally evict CtBP from occupied promoter regions of target genes while NSC95397 is a newly identified compound as a potential inhibitor of CtBP interaction with ZEB1[29,42]. The inhibition of ZEB1–CtBP would de-repress expression of genes like *CDH1* and CDK inhibitors thereby possibly inhibit cell proliferation. We used 10 μM NSC95397 and 10 mM MTOB based on reports[29,42] to modulate Zeb1 activities both in vitro and in vivo. mRMVEC immunostaining provided evidence that Zeb1 was clearly present not only in the nucleus but also in the cytosol of about 30% cells (Fig. 5a, b). However, an addition of the ZEB1–CtBP inhibitor NSC95397 to the culture medium significantly relocated Zeb1 from the nucleus to the cytosol in almost 100% cells (Fig. 5a, b), whereas the inhibitor MTOB had little effect on the translocation of Zeb1 (Fig. 5a, b). By contrast, the treatments of both NSC95397 and MTOB did not affect the nuclear location of Ctbp (Fig. 5a, b). To validate the relocation of Zeb1 characterized by immunofluorescence ((IF); Fig. 5a), we isolated both nuclear and cytosolic fractions of total protein samples from both PBS control mRMVECs and cells treated with either MTOB or NSC95397 for Zeb1-probed WB. We also used Ctbp and Actb as relevant nuclear and cytosolic controls though Actb also showed high expression in the nucleus as reported[43] (Fig. 5c). The treatment with 10 mM MTOB had no significant effect on the relocation of Zeb1 from the nucleus to the cytoplasm whereas the treatment with 10 μM NSC95397, however, significantly increased cytosolic Zeb1 compared to the PBS control (Fig. 5c), which confirms the relocation of Zeb1 upon the ZEB1–CtBP inhibitor NSC95397 treatment.

Interestingly, both ZEB1–CtBP inhibitors significantly reduced *Zeb1* expression though 10 μM NSC95397 was much more effective than 10 mM MTOB (Fig. 5d). Again, the reduction of Zeb1 expression by both ZEB1–CtBP inhibitors did not decrease *Vegfa* expression in cultured mRMVECs (Fig. 5d). As a result, the downregulation of *Zeb1* by the Zeb1 inhibitors reduced mRMVEC proliferation in culture (Fig. 5d, e). Ten micromolar NSC95397 and 10 mM MTOB also reduced the capacities of mRMVEC migration (Fig. 5f and Supplementary Fig. 1a) and tube formation (Fig. 5g and Supplementary Fig. 1b) in vitro. Taken together, we conclude that the functional disruption of Zeb1–Ctbp complex by the ZEB1 inhibitors is realized through both the translocation of Zeb1 to the cytosol from the nucleus where it serves as a transcription factor to regulate expression of target genes, and the attenuation of Zeb1 that results in the inhibition of mRMVEC proliferation, migration, and tube formation.

To check whether or not downregulation of Ctbp1 and 2 affect Zeb1 expression and/or relocation, we manufactured Ctbp1 and 2 shRNA lentivirus particles released in the culture medium using a commercial mixture of Ctbp1 and 2 shRNA lentiviral vector

plasmids (see "Methods"). We utilized Ctbp1 and 2 shRNA lentivirus to knockdown Ctbp1 and 2 in mRMVECs (Ctbp_sh). About 70−80% downregulation of Ctbp1 and 2 in mRMVECs by the Ctbp1 and 2 shRNA lentivirus was validated by WB using an antibody against both Ctbp1 and 2 (Fig. 6a) as both CtBP1 and CtBP2 interact with ZEB1 and have a similar biological function as a co-repressor in the nucleus[44]. Apparently, a large quantity of Ctbp was present in the nucleus compared to a relatively small amount of Zeb1 (Fig. 6a) and downregulation of Ctbp1 and 2 did not significantly affect the Zeb1 expression detected by the WB (Fig. 6a), Zeb1 relocation (Fig. 6b, c), and cell proliferation (Fig. 6d). These results indicate that compared to Zeb1 that exists in both the nucleus and cytoplasm, Ctbp is exclusively present in the nucleus (Fig. 5a, c), and that in the nucleus Zeb1 mostly interacts with Ctbp to repress expression of genes involved in cell proliferation in mRMVECs.

**Zeb1 appears to regulate itself through the miR-200 family.** It was surprising that the ZEB1–CtBP inhibitors significantly reduced *Zeb1* expression in mRMVECs because based on our understanding, the inhibitors are not supposed to affect *Zeb1* gene expression (mRNA levels). It is possible that these ZEB1–CtBP inhibitors remove and/or prevent the repressive Ctpb from interacting with Zeb1 so that other promotive partners like P300 would fit in to switch Zeb1 from repressing to promoting expression of target genes such as the *miR-200* family, resulting in repression of *Zeb1*. To verify whether the ZEB1–CtBP inhibitors functionally disrupt the Zeb1–Ctbp complex, using a CtBP anti-body we co-immunoprecipitated Zeb1 from total protein extracts of mRMVEC cells treated with the inhibitor NSC95397 or MTOB. As expected, the amounts of Zeb1 detected by a Western blot (WB) were significantly less in the ZEB1–CtBP inhibitor-treated cells compared to the PBS-treated control cells (Fig. 6e). NSC95397 almost completely disrupted Zeb1–Ctbp complex whereas MTOB only marginally finished the job (Fig. 6e), which explains why NSC95297 was much more effective than MTOB in inhibiting cell proliferation (Fig. 5e), cell migration (Fig. 5f), and cell tube formation (Fig. 5g). The disruption of the Zeb1–Ctbp complex may therefore de-repress the miR-200 family, a group of well-known intracellular repressors of ZEB1, and thereby repress Zeb1 expression[15]. To check whether inhibition of Zeb1 by MTOB and NSC95397 was through upregulation of the *miR-200* family members, we performed a qPCR to check the amounts of their messages in mRMVECs. As expected, the ZEB1–CtBP inhibitors significantly increased the expression of the *miR-200* family members, particularly *miR-200b* and *miR-200c* (Fig. 6f). In addition, we had previously demonstrated that Zeb1 physically binds to the promoters of *miR-200a/b/c*[18]. These evidences demonstrate that ZEB1–CtBP inhibitors inactivate the repressive Zeb1–Ctbp complex by eviction of Ctbp and thereby depression of the *miR-200* family, leading to the downregulation of *Zeb1* (Fig. 5d).

**Topic application of ZEB1–CtBP inhibitors reduces corneal NV.** As demonstrated above, the ZEB1–CtBP inhibitors caused mRMVEC senescence in culture (Fig. 5e) likely through Zeb1 reduction-induced augmentation of the cell Cdk inhibitors (Fig. 5d). To test whether ZEB1–CtBP inhibitors can be used to treat corneal NV, we topically applied 2.5 μl of 10 μM of NSC95397 or 10 mM MTOB to the eyes with the alkali-induced corneal NV twice a day for two weeks. The NV severity of the NSC95397-treated eyes was significantly reduced compared to the PBS-treated control eyes, whereas no significant difference was observed between the MTOB-treated eyes and the PBS-treated control eyes (Fig. 6g–i). This in vivo experiment result was

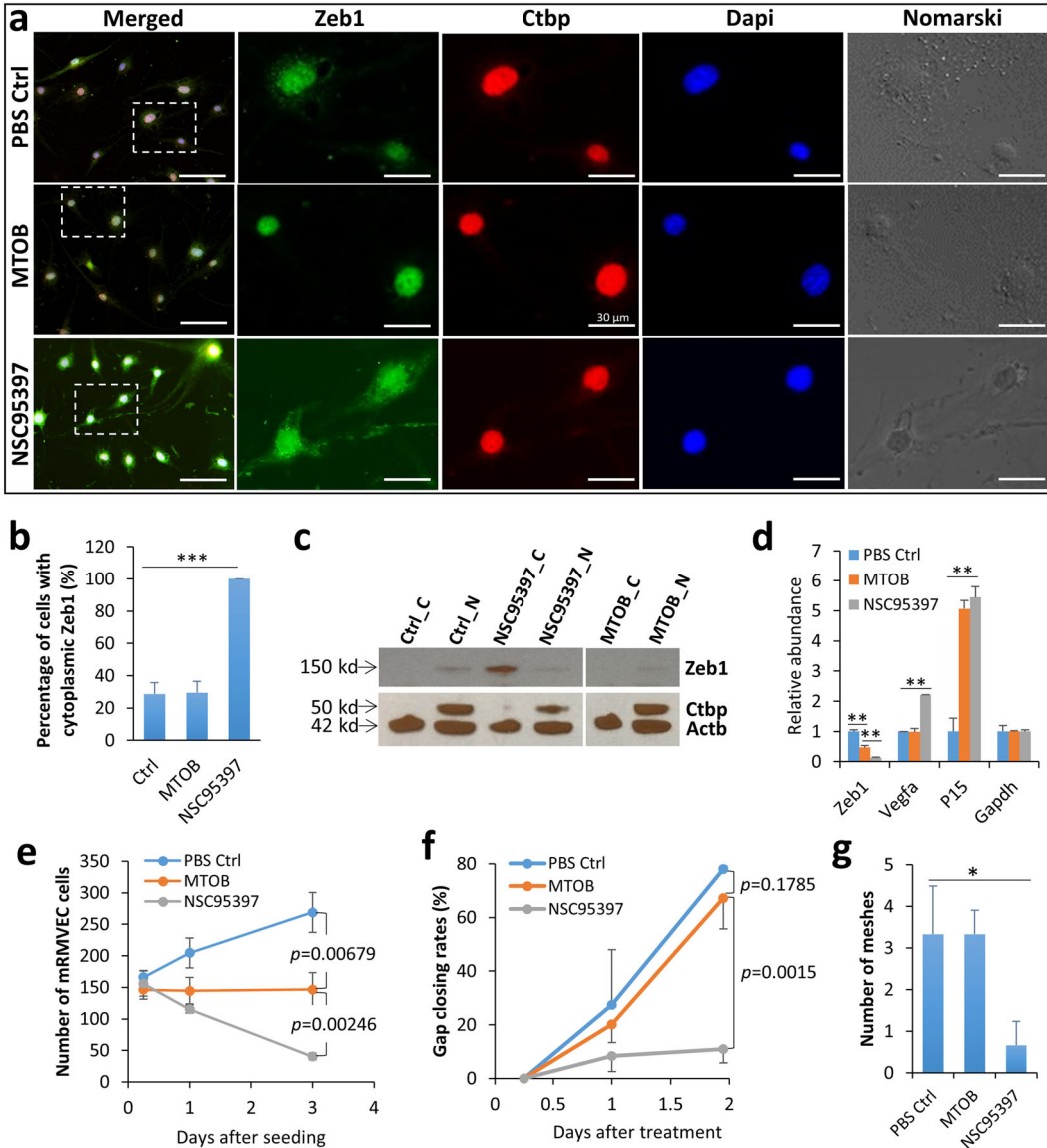

**Fig. 5 Treatment with the ZEB1–CtBP inhibitor NSC95397 causes translocation and reduction of Zeb1, proliferation, migration, and tube formation of mRMVECs.** Compared to PBS control, (**a**–**b**) one-day 10 μM NSC95397 treatment translocated Zeb1 from the nucleus to the cytosol in mRMVECs while 10 mM MTOB had no such an effect. **c** This Zeb1 translocation was validated by WB on nuclear and cytosolic fractions of total protein samples isolated from the PBS control, NSC95397- or MTOB-treated mRMVECs using Ctbp and Actb antibodies as relevant nuclear and cytosolic controls though Actb also showed high expression in the nucleus as reported[43]. "C" for cytosolic fraction while "N" for nuclear fraction. **d** Both 10 mM MTOB and 10 μM NSC95397 significantly reduced *Zeb1* mRNA in mRMVECs and **e** cell proliferation rates, but only 10 μM NSC95397 significantly reduced (**f**) cell migration, and **g** cell tube formation. *$p \leq 0.05$; **$p \leq 0.01$. Scale bars represent 100 μm.

consistent with the in vitro mRMVEC assessments in which NSC95397 was much more efficient to inhibit cell proliferation, migration, and tube formation than MTOB even though the concentration of MTOB was 1000 times higher than that of NSC95397 (Fig. 5e–g). Taken together, we conclude that NSC95397 is more effective than MTOB in reducing Zeb1 expression and vascular EC proliferation, migration, and tube formation and it possesses a therapeutic potential in future clinical applications.

## Discussion

The progression of NV depends on vascular EC proliferation; in addition to an increase in expression of VEGF for induction of NV, keeping EC division after the initial NV induction is critical to accomplish the extension and maintenance of new blood vessels[7,45]. ZEB1 is one of the important factors directly involved in driving tumor cell proliferation through repressing CDK inhibitors[40]. However, it was not known whether ZEB1 plays an important role in regulation of vascular EC proliferation in angiogenesis. Here, we provide evidence that Zeb1 is required for the lung capillary development in mouse embryos. We also show that Zeb1 facilitates the development of mouse alkali-induced corneal NV. This is an important discovery as the most attention so far is paid to those promoting and inhibitory cytokines for NV, particularly VEGF. VEGF antibodies have been widely used as first line therapeutics to treat neovascular age-related macular degeneration (nAMD) in the clinic. Although the intraocular anti-VEGF therapy has had clinical success, many nAMD patients do not attain significant visual improvement. For examples, 50–80% of nAMD patients never achieve 20/40 vision

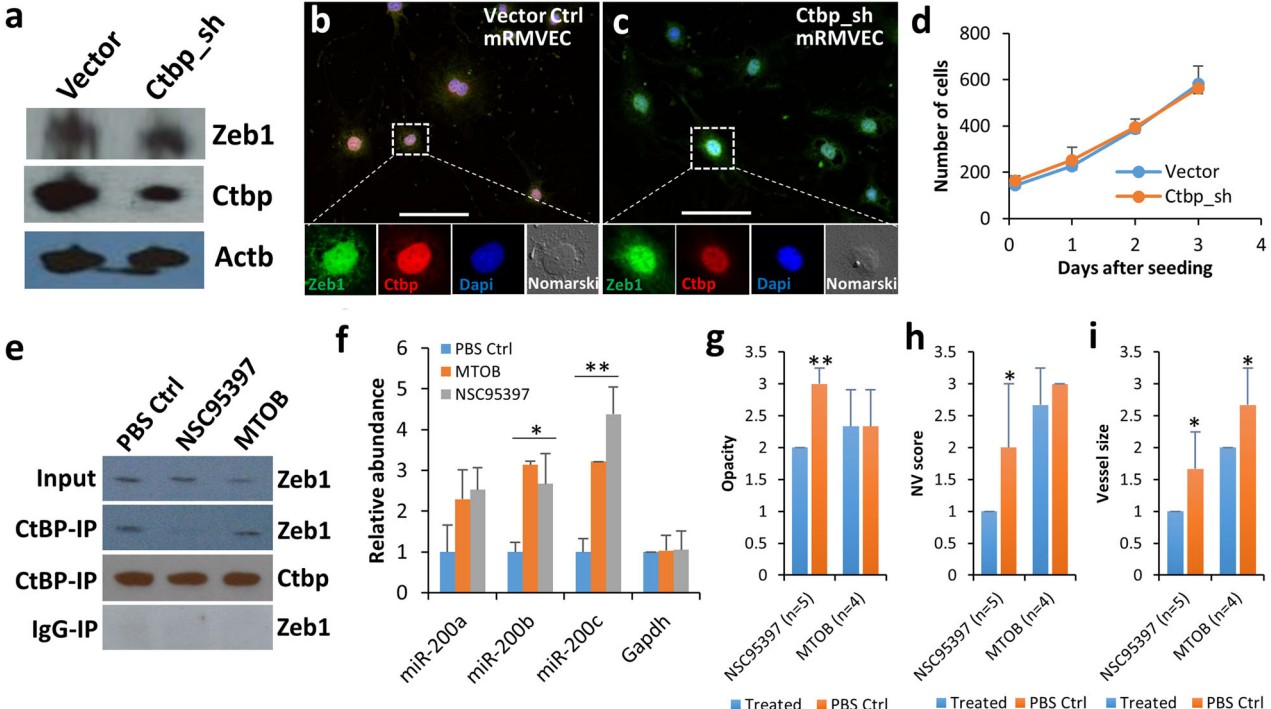

**Fig. 6 The ZEB1–CtBP inhibitor NSC95397 evicts Ctbp from Zeb1 complex and thereby upregulates the miR-200 family to downregulate Zeb1 expression in mRMEVCs.** Downregulation of Ctbp by lentiviral shRNA in mRMEVCs did not affect **a** expression and (**b**–**c**) relocation of Zeb1, and thereby (**d**) cell proliferation. **e** Co-immunoprecipitation of Zeb1 using a CtBP antibody against both CtBP1 and 2 (CtBP-IP) from total protein (Input) isolated from mRMVECs treated with either 10 μM NSC95397 or 10 mM MTOB and WB for Zeb1 and Ctbp. **f** Expression of the *miR-200* family genes detected by qPCR in mRMVECs treated with 10 mM MTOB or 10 μM NSC95397. It appears that only 10 μM NSC95397 significantly reduced the alkali-induced corneal NV evaluated by criteria of (**g**) opacity, (**h**) NV score, and **i** vessel size. *$p \leq 0.05$; **$p \leq 0.01$. Scale bars represent 100 μm.

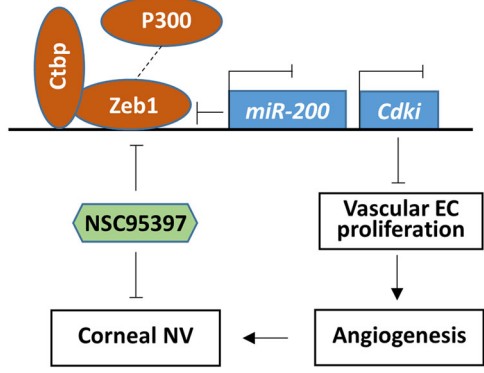

**Fig. 7 Schematic diagram of Zeb1-regulation of corneal neovascularization (NV).** The repressive complex of Zeb1-Ctbp sits on the promoters of target genes, including the *miR-200* family and cyclin-dependent kinase inhibitors (*Cdki*) and represses their expression. The ZEB1–CtBP inhibitors such as NSC95397 evict Ctbp from the complex and let new interacting partners like P300 fit in to form a promotive complex to induce expression of both the *miR-200* family and *Cdki*. The miR-200 family in turn repress Zeb1 expression while Cdki block vascular EC proliferation and thereby angiogenesis, leading to reduction of the alkali-induced corneal NV.

and 12–25% have 20/200 vision or worse despite the treatment[11,46]. Critically, of those whose vision improve following treatment, gain in visual acuity are usually lost within 4 years, and nearly all (98%) develop untreatable central retinal atrophy with 7 years[10,46]. Adverse effects of the VEGF neutralization on multiple retinal cell types, widely reported in animal models, are observed in patients who have been treated with anti-VEGF drugs for

several years[10]. It has just started to apply the anti-VEGF therapy to treat corneal NV though with a risk of reduction of corneal epithelium repair[3,47]. A new strategy based on different mechanisms of NV is needed to explore new avenues for treating ocular NV diseases. Reducing ZEB1 expression may be one of such avenues.

We have demonstrated that Zeb1 promotes angiogenesis through the repression of Cdk inhibitors to promote vascular EC proliferation; and the reduction of its expression by the loss-of-function mutation, by the knockdown of mRNA, and by the inhibition of its interaction with Ctbp would negatively affect angiogenesis (Fig. 7). However, whether Zeb1-upregulation of cell proliferation is specific to the vascular vessel endothelium is so far not clear. In the cornea, there are multiple cell types including epithelial cells, stromal keratocytes and resident immune cells, and corneal endothelial cells. As regards to the alkali-induced corneal NV, the topic application of the alkali immediately damages the corneal epithelium and underneath tissues, and causes inflammation in the stroma, leading to corneal edema and NV[48]. In the quiescent cornea, Zeb1 was present in a large quantity in the epithelium basal cells, but mostly in the cytosol while the amounts of Zeb1 did not positively correlate with the proliferative marker Ki67 in the nucleus (Supplementary Fig. 2a). The alkali-caused corneal wound increased Zeb1 expression not only in the epithelial basal cells but also in the stromal keratocytes, positively correlating with the increase in Ki67+ cells in both tissues (Supplementary Fig. 2b) compared to the PBS control cornea (Supplementary Fig. 2a). Intriguingly, the inhibition of Zeb1–Ctbp interaction by the topical application of the inhibitor NSC95397 to the alkali-burned mouse corneas seemingly did not affect the epithelium wound healing, but did inhibit the NV (Fig. 6g–i) likely by reducing Zeb1 in vascular ECs in the cornea

(Fig. 3). The reason(s) for why the topical application of the ZEB1–CtBP inhibitor did not affect corneal epithelium repair is not known. One possibility is that corneal epithelial cells are less sensitive to the inhibitor than other corneal stromal cells because they express high amounts of Zeb1 (Fig. 3 and Supplementary Fig. 2) and the reduction of Zeb1 by the inhibitor does not reach the low threshold required for the epithelial cell proliferation. It is of note that the small molecule NSC95397 exerts multiple inhibitory function in addition to interrupt the CtBP–ZEB1 interaction, it has been reported to disrupts the S100A4/myosin-IIA interaction and inhibits S100A4-mediated depolymerization of myosin-IIA filaments. S100A4, also known as fibroblast-specific protein 1 (FSP1) is a typical marker for mesenchymal cells. Repression of ZEB1 often leads to repression of S100A4. Clearly, NSC95397 is an important small molecule that inhibits EMT through inhibition of both S100A4/myosin-IIA and CtBP–ZEB1 interaction. In addition, S100A4 was also reported to be an angiogenic factor, suggesting that NSC95397 is an antiangiogenic substance by directly inhibiting both S100A4 and ZEB1 functions.

Wounding-affected corneal cells secret large amounts of cytokines, including TGFβ that can upregulate Zeb1[21,49,50]. TGFβ binds to type I and II serine/threonine kinase receptors and phosphorylates them. The phosphorylation of these kinase receptors activates Smad2 and Smad3. The activated Smad2/3 form complexes with Smad4, and translocate into the nucleus where the Smad complexes interact with various transcription factors and transcriptional co-activators, and regulate the transcription of target genes such ID2 and ETS1[51]. ETS1 acts in cooperative fashion with E2A proteins released from ID proteins, and is involved in upregulation of ZEB1[51]. There was no evidence to support that ZEB1 directly upregulates angiogenic cytokines like VEGF though some evidences did show that ZEB1 might upregulate VEGF indirectly through repression of the miR-200 family that represses VEGF[28]. We showed that the disruption of Zeb1 interaction with Ctbp by the ZEB1–CtBP inhibitors did upregulate the miR-200 family members (Fig. 6f), but did not repress *Vegf* expression (Fig. 5d). Stressed corneal epithelial cells secret interleukin (IL)-1β and tumor necrosis factor alpha (TNFα)[8]. Both IL-1β and TNFα can cause increased infiltration of immune cells into the cornea, leading to NV[8,50]. We showed that the alkali-induced corneal wound could increase Zeb1 expression (Fig. 3 and Supplementary Fig. S2) while the depletion of Zeb1 clearly decreased TNFα expression though increased IL-1β expression in MEFs (Fig. 4b). However, TNFα has been reported to counteract against both TGFβ and VEGF, thereby reduces angiogenesis in the cornea[52]. There are too many known and unknown cytokines that might be involved in promoting angiogenesis, it is an ongoing effort to clarify whether ZEB1 has other indirect role in regulation of angiogenesis.

## Methods

**Animals and alkali-induced corneal NV.** Young (6–8 weeks) female Zeb1[+/−] and Zeb1[+/+] and C57BL/6J mice were anesthetized by an intraperitoneal (IP) injection of 100 mg/kg ketamine and 5 mg/kg xylazine and the eyelashes were trimmed. The alkali-induced corneal NV model was created by placing a 3 M filter disc of 2 mm diameter soaked with 2.5 μl of 1 N NaOH on the center surface of the cornea for 10 s followed by a thorough rinse with PBS for 15 s. For Zeb1 influence assessment, five Zeb1 heterozygous knockout (Zeb1[−/+]) and five wild-type (Zeb1[+/+]) mice were used. For ZEB1–CtBP inhibitor assessment, the C57BL/6J mice were divided into three groups of five mice. Topical application of 2.5 μl of PBS (group 1), or 10 mM MTOB (group 2), or 10 μM NSC95397 (group 3) twice a day for 2 weeks to the cornea[53]. The evaluation of corneal NV was conducted in 2 weeks after the alkali treatment[53] by two blinded and well-trained persons under a stereoscope. The evaluation was based on three criteria: opacity (scale 0–3), NV score (scale 0–3), and vessel size (scale 0–3)[53]. All aspects of this study were conducted in accordance with the policies and guidelines set forth by the Institutional Animal Care and Use Committee and were approved by the University of Louisville, Kentucky, USA.

**Cell lines and culture.** Two kinds of cells were used for in vitro experiments. Mouse embryonic fibroblasts (MEFs) were prepared from E17.5 Zeb1[+/+], Zeb1[+/−], and Zeb1[−/−] embryos[40], whereas mouse retinal microvascular endothelial cells (mRMVECs) were purchased from Cell Biologics (Cat. #: C57–6065). They were cultured in DMEM medium with 10% fetal bovine serum (FBS) and Endothelial Cell Medium (Cell Biologics cat. #: M1168), respectively, under 5% $CO_2$ at 37 °C. The above media were mixed with 1% penicillin and streptomycin antibiotics and refreshed every 3 days until cell confluence.

**Cell migration assay.** Monolayer-cultured cells at confluence were treated with 5 μg/ml mitomycin C for 2 h at 37 °C and then washed with PBS, followed by a straight scratch using a 200P pipette tip, and pictured under an inverted microscope and re-pictured every day at the same location. Width of three scratched gaps for each treatment was measured every day and the invert of the measurement was served as a gap closing rate.

**Cell tube formation assay.** An aliquot of Matrigel (BD Bioscience) was warmed up at room temperature and then left on ice briefly. Hundred-fifty microliters of Matrigel was plated to a 48-well plate at a horizontal level that allows the Matrigel to distribute evenly, and incubated for 30 min at 37 °C. mRMVECs ($9 \times 10^4$) were resuspended with serum-free Endothelial Cell Medium, and loaded on the top of the Matrigel. Following incubation at 37 °C overnight, tubules (meshes) in each field were pictured and an average of tubules from 3 to 5 random fields in each well was counted.

**Cell immunofluorescence (IF).** Cells were adherently cultured in 8-well chamber glass slides coated with 0.1% gelatin for 2 days or as otherwise specified. They were fixed with 4% paraformaldehyde (PFA), rinsed with PBS, and blocked with 1% bovine serum albumin (BSA) and 3% serum of the species in which the secondary antibody was raised. Rabbit polyclonal Zeb1 antiserum (1:500, a gift from Dr. Douglas Darling) and mouse monoclonal CtBP antibody (1:200, Santa Cruz cat. #: sc-17759) were used as primary antibodies together with the secondary antibody Alexa Fluor 488 goat anti-mouse-IgG (1:500, ThermoFisher cat. #: A32723) or Alexa Fluor 594 goat anti-rabbit-IgG (1:500, ThermoFisher cat. # A32740). Nuclei were counterstained with the Hoechst dye (1:500, Invitrogen cat. #: H1399), and images were captured with a Zeiss fluorescence microscope.

**Corneal flat-mount immunostaining.** The enucleated eye was fixed in 4% PFA for 4 h and rinsed with PBS. The entire cornea was removed with the limbus from the posterior portion of the eye and then cut into four quadrants of approximately equal size using a pair of surgical scissors under a surgical microscope and then lied flat on a slide. The flat corneal tissue went through following steps: blocking with 1% BSA and 3% serum, hybridizing with the first mouse antibody CD31 (1:50, BD cat. #: 550274) and the rabbit antibody LYVE-1 (1:500, Novus cat. #: NB600-1008) in the blocking buffer and then the secondary antibodies as with the above IF. Nuclear counterstaining and photography were as with the above IF.

**Immunohistochemistry (IHC).** Eyeballs from PBS-sham-treated control and the CtBP–ZEB1 inhibitor-treated mice were frozen in optimal cutting temperature (OCT) medium for at least 4 h. The frozen tissues were cryosectioned for IHC with the primary antibodies Zeb1 (1:200, a gift from Dr. Douglas Darling), CD31 (1:50, BD cat. #: 550274), and Ki67 (1:50, BD cat. #: 550609) and processed as above IF.

**Real-time quantitative PCR (qPCR).** Total RNA from cells or corneal tissues was extracted using TRIzol solution (Invitrogen), and complementary DNA (cDNA) of mRNA was synthesized using the Invitrogen RT kit (Invitrogen), whereas cDNA of microRNA (miR) was prepared as previously described[18,54]. Briefly, the polyadenylation of at least 5 μg of the total RNA was completed by the poly(A) polymerase kit (PAP; Ambion) in a 20-μl reaction volume according to the manufacturer's instructions. The polyadenylated miR was thereafter utilized directly for miR cDNA preparation using the reverse transcription kit (M-MLV reverse transcriptase; Invitrogen) and the adaptor primer (5′-GCGAGCACAGAA TTAATACGACTCACTATAGG(T)12VN*−3′) in a 40-μl reaction volume. qPCR was performed using a universal primer (5′-GCGAGCACAGAATTAATACGA C-3′) and a miR-specific primer (Supplementary Table 1). SYBR Green (Molecular Probes) qPCR was performed using the Stratagene Mx3000P system. Regular PCR primer sets were designed using the website-based program Primer3; their sequences are listed in Supplementary Table 1. The relative amounts of the target mRNA were estimated by the threshold cycle (Ct) values using the double-delta method ($2^{-(\Delta Ct_1 - \Delta Ct_2)} = 2^{-\Delta\Delta Ct}$, where sample 1 is compared to sample 2)[55] and normalized to the levels of the housekeeping gene *Gapdh*. At least three biological samples were analyzed, each in duplicate.

**Lentiviral shRNA.** The target sequence of the short-hairpin RNA (shRNA) oligomer used for Zeb1 silencing was 5′- AAGACAACGTGAAAGACAA (Zeb1_sh). The lentiviral vector construction (Vector Ctrl) and virus assembly procedures were as described previously[56]. A mixture of 3 Ctbp1 (Santa Cruz Biotechnology, cat. #: sc-35121-SH) and 3 Ctbp2 shRNA lentiviral vectors with puromycin

selection (Santa Cruz Biotechnology, cat. #: sc-37768-SH) were purchased and their lentivirus particles were manufactured using 293T cells and Lipofectamin3000 (Invitrogen, cat. #: L3000-015) according to the manufacturer's instruction. Briefly, 293T cells were cultured until 70–80% confluence when the mixture of lentiviral package plasmids (pMDLg/pRRE, pRSV.Rev, and pMD2.G at ratio of 1:1:1) were further mixed with the above Ctbp1 and Ctbp2 shRNA lentiviral vectors at 1:1 ratio. OPTI-MEM and lipofectamin3000 were sequentially added into a tube containing the above plasmids and incubated at RT for 10 min. The whole mixture was thereafter carefully added to the 293T cells cultured with fresh DMEM plus 10% FBS. Two days late collect the medium that contains the lentiviral particles for 3 days. Primary mRMVECs were infected in culture by Zeb1_sh, Ctbp_sh, and Vector Ctrl lentivirus. The transduction efficiency for Zeb1_sh was >70% based on EGFP-positive cell counts, whereas Ctbp-sh transduced mRMVECs were further selected in culture by 5 μg/ml puromycin for two more weeks. The downregulation of Zeb1 and Ctbp was quantitatively analyzed by qPCR (Fig. 4f) and WB (Fig. 6a), respectively.

**Co-Immunoprecipitation, nuclear and cytosolic protein fractionation, and western blotting (WB).** mRMVECs treated with 10 mM 4-methylthio 2-oxo-butyric acid (MTOB) (Sigma cat. #: K6000), or 10 μM 2,3-bis{[(2-Hydroxyethyl) thiol]}−1,4-naphthoquinone (NSC95397) (Sigma cat. #: N1786), or PBS, were lysed in protein extraction lysis buffer (RIPA buffer: 150 mM, 5 mM EDTA, 50 mM Tris-HCl pH 8.0, 1% NP-40) on ice for 20 min, followed by a 10-min centrifugation at 13,000 r.p.m. The supernatant was mixed with either the CtBP mouse monoclonal antibody attached with protein A/G agarose beads (Santa Cruz cat. #: sc-17759AC) against both Ctbp1 and 2 or normal mouse IgG (Santa Cruz cat. #: sc-2343AC) at the 1:10 ratio of total protein volume on a horizontal shaker for 10 min at 4 °C. To separate total protein into nuclear and cytosolic fractions, mRMVECs were lysed in HEPES buffer (10 mM HEPES, pH7.5, 10 mM KCl, 0.1 mM EDTA, mM dithiothreitol, 0.5% NP-40, 0.5 mM PMSF, Sigma proteinase inhibitor cocktail) and centrifuged at 12,000 × $g$ at 4 °C for 10 min. The supernatant was saved as a cytosolic fraction. The nuclear pellet was resuspended in nuclear extraction buffer (20 mM HEPES pH7.5, 400 mM NaCl, 1 mM EDTA, 1 mM dithiothreitol, 1 mM PMSF, Sigma proteinase inhibitor cocktail) for 30 min on ice with vortexing at 10 min internals and then centrifuged at 12,000 × $g$ for 30 min. The supernatant now was saved as a nuclear fraction. The amounts of total protein were quantified. A 4–21% gradient sodium dodecyl sulfate–polyacrylamide gel electrophoresis gel was loaded with 10 μg of the above protein samples, and an electrophoresis was performed at 120 V for 60 min. The proteins in the gel were transferred to a polyvinylidene chloride membrane at 4 °C overnight. The protein membrane was incubated with the Zeb1 antiserum (Douglas Darling's gift), CtBP antibody (Santa Cruz cat. #: sc-17759), and the secondary antibody (anti-rabbit IgG HRP, Santa Cruz cat. #: sc-2004) in blocking solution. The amounts of Zeb1, Ctbp, and Actb on the membrane were visualized with the Amersham ECL kit (Cat. #: RPN 2106) and detected by an X-ray film.

**Statistics and reproducibility.** Where applicable, data were analyzed by the two-tail unpaired $t$-test. Values in the graphs were presented as means ± standard deviations. Three-star "***" indicates $p$-value ≤ 0.001, two-star "**" indicates $p$-value ≤ 0.01, whereas one-star "*" indicates $p$-value ≤ 0.05. For all studies, results were obtained from at least three biological samples (animals and cultured cells), unless otherwise specified.

**Reporting summary.** Further information on research design is available in the Nature Research Reporting Summary linked to this article.

## Data availability

Raw data including animal ID, cell biological replicates, numerical calculation, and statistical analyses are detailed in Supplementary Data 1. Remaining information including real-time qPCR analyses can be provided from the corresponding author upon reasonable request,

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

## Acknowledgements

This work was supported by the Natural Science Foundation of Liaoling Province (2019-MS-068 to L.J., 201602210 and 20180550976 to L.Z.), National Institute of Health (EY024110 to D.C.D., P20GM103453 to Y.L.), Basic Research Grant of University of Louisville School of Medicine (E0819 to Y.L.), and Research to Prevent Blindness (to DOVS). The funders had no role in study design, data collection, data analysis, interpretation, writing of the report. We also thank Dr. Chi Li at James Brown Cancer Center, University of Louisville School of Medicine for his help in manufacturing Ctbp shRNA lentivirus.

## Author contributions

L.G., Y.N., and W.L. performed the experiments, collected, and analyzed the data. X.L. conducted Zeb1-knockout mouse genotyping and finished part of the immunostaining work. L.P. assisted in isolation of mouse corneas. W.W. and H.J.K. helped designing the experiments and analyzing the data. D.C.D., L.Z., and Y.L. designed experiments, analyzed the data, and wrote the manuscript. All authors read and approved the final manuscript.

## Competing interests

The authors declare no competing interests.
