## [Peer Review File · Communications Biology]

Reviewers' comments:

Reviewer #1 (Remarks to the Author):

Overview

In this manuscript, Liu et al examine the role of ZEB1 in the formation of corneal neovascularization (CNV) using in vitro and in vivo models. This study is an obvious area of interest and importance among angiogenesis, transcriptional, and eye biologists. The potential future therapeutic application is also intriguing given that current VEGF inhibitors can fail due to VEGF-independent CNV. Indeed, the results reveal that the transcription factor ZEB1 plays a critical role in CNV formation. As such, the authors include studies with MTOB and NSC95397 small molecule CtBP inhibitors to study the role of CtBP-ZEB1 interaction in ZEB1 transcription regulation, as direct ZEB1 inhibitors are not available, and they find that CtBP inhibition evicts ZEB1 from the nucleus and lowers its levels, contributing to amelioration of parameters associated with CNV in vitro and in vivo. Though of potential broad interest to the fields of ophthalmology and angiogenesis, the manuscript in current form suffers from a lack of direct validation of the role of CtBP in ZEB1 regulation of CNV due to the sole reliance on inhibitor studies without inclusion of genetic validation as with RNAi techniques. For this and other technical reasons below, the manuscript in its current form is not suitable for publication in Communications Biology. Major/minor critiques and suggested revisions are noted below.

Major critiques:

- 1) Fig. 5A: It is impossible to identify cytosol and nucleus in the submitted images in Fig. 5A. Higher magnification and resolution is necessary, along with statistically valid quantitative analysis of the relocation phenomenon. In addition, the authors need to utilize an independent method to characterize the translocation- this could be to isolate nuclear and cytosolic extract and then blot for the levels of Zeb1 with relevant nuclear and cytoplasmic controls for purity of the fractionation.
- 2) The authors should use IF or fractionation to test if CtBP (and specifically CtBP1 vs. CtBP2 as their localization can differ) co-relocalizes with ZEB1 after inhibitor treatment.
- 3) The authors must use genetic knockdown of CtBP1, 2, or both to substantiate their claim that CtBP inactivation relocalizes ZEB1 and alters ZEB1 level and function in CNV associated phenomena in vitro- either by si or sh techniques, including rescue of effects with rescue CtBP1/2 expression, as the specificity of the compounds used has not been absolutely established in this cell system.
- 4) The utility of MTOB is generally limited as it is not an "inhibitor" of CtBP- it is CtBP's highest affinity dehydrogenase substrate (see Ref # 42)- and acts as an inhibitor at extremely high concentrations by virtue of substrate inhibition. Newer generation CtBP dehydrogenase inhibitors have been reported (for example: see Korwar et al. Bioorg Med Chem. 2016 24:2707 describing "HIPP"-class inhibitors) which may be more effective and better complement the NSC experiments in vitro and in vivo by showing efficacy of an entirely different class of CtBP inhibitor.
- 5) Fig. 6A: The co-IP experiment lacks controls and blotting for the IP components. An IgG control IP should be shown as a negative control that the ZEB1 band seen is due to specific IP. Input lysates used in the IP's, blotting for both ZEB1 and CtBP should be shown. The CtBP blot for the IPs should be shown. Comment in the results that the CtBP antibody used recognizes both CtBP1 and CtBP2, and justify why it is not necessary to know which CtBP is binding to ZEB1 in these cells.
- 6) Although the authors show evidence that NSC95397 acts by evicting CtBP from Zeb1, the molecule also inhibits other proteins, namely S100a4 (Biochemistry 2011;50:7218), which also acts as an angiogenic factor (Oncogene. 2001; 20:4685). The authors should comment in the Discussion on the potential impact of this effect of NSC95397 on an angiogenic factor vs. its effect on the the angiogenic CtBP/ZEB1 interaction.

Minor critiques:

- 1) Line 121: Please cite the appropriate reference for ZEB1 knockout embryonic lethality.
- 2) Fig. 5D. Typographical error noted in Y-axis legend.

Reviewer #2 (Remarks to the Author):

In this study, Lei Jin and colleagues demonstrated an alternative mechanism for Zeb1-regulation of corneal neovascularization. The authors showed that Zeb1 functioned as an important factor to promote vascular EC proliferation and corneal neovascularization in a VEGF-independent manner in vitro and in vivo. Moreover, treatment with the small molecule inhibitors of CtBP, including MTOB and NSC95397, resulted in dissociation of the Zeb1-Ctbp complex and consequent upregulation of the miR-200 family, leading to inhibition in Zeb1 expression, mRMVEC proliferation and mouse corneal NV severity. However, the authors did not provide enough solid experimental evidences that can support their conclusion. The manuscript would be further improved by addressing the following issues:

1. The authors used numerous in vitro and in vivo models in their study, such as Zeb1 knockout mouse embryo lungs in Figure 1 and Zeb1-depleted MEFs in Figure 4. However, based on these results, it is hard to evaluate the role of Zeb1 in corneal ECs and neovascularization.
2. In Figure 1, the authors should explain how the capillary ECs and mesenchymal cells were examined and calculated in lung tissue sections just based on the H&E staining results.
3. In Figure 3, according to IF staining, it is hard to conclude that Zeb1 expression was specifically upregulated in ECs of newly formed blood vessels in the alkali-burned mouse corneas. The results should be quantified.
4. In Figure 4B and F, depletion of Zeb1 resulted in significant upregulation of Vegf/Vegfr expression in MEFs and mRMVECs. Why the authors suggested that Zeb1-regulation of corneal neovascularization is independent of Vegf pathway? This should be addressed.
5. In Figure 5A, the expression of Zeb1 at the protein level and cell proliferation in mRMVECs was not significantly inhibited by treatment with MTOB or NSC95397. However, the authors showed decreased expression of Zeb1 at the mRNA level in Figure 5B and reduced cell proliferation in Figure 5C. This are confused and should be further evaluated.
6. In Figure 6A, the expression of Zeb1 and CtBP in the input and Co-IPs for IgG and anti-CtBP should be examined.
7. Considering that Zeb1 and the miR-200 family members are regulated in a negative feedback loop, it is not appropriate to simply conclude that inhibition of Zeb1 expression by MTOB and NSC95397 was mediated by the upregulation of miR-200 family members in Figure 6 and 7. This should be further addressed.

Reviewer #1 (Remarks to the Author):

In this manuscript, Liu et al examine the role of ZEB1 in the formation of corneal neovascularization (CNV) using in vitro and in vivo models. This study is an obvious area of interest and importance among angiogenesis, transcriptional, and eye biologists. The potential future therapeutic application is also intriguing given that current VEGF inhibitors can fail due to VEGF-independent CNV. Indeed, the results reveal that the transcription factor ZEB1 plays a critical role in CNV formation. As such, the authors include studies with MTOB and NSC95397 small molecule CtBP inhibitors to study the role of CtBP-ZEB1 interaction in ZEB1 transcription regulation, as direct ZEB1 inhibitors are not available, and they find that CtBP inhibition evicts ZEB1 from the nucleus and lowers its levels, contributing to amelioration of parameters associated with CNV in vitro and in vivo. Though of potential broad interest to the fields of ophthalmology and angiogenesis, the manuscript in current form suffers from a lack of direct validation of the role of CtBP in ZEB1 regulation of CNV due to the sole reliance on inhibitor studies without inclusion of genetic validation as with RNAi techniques. For this and other technical reasons below, the manuscript in its current form is not suitable for publication in Communications Biology. Major/minor critiques and suggested revisions are noted below.

Major critiques:

1) Fig. 5A: It is impossible to identify cytosol and nucleus in the submitted images in Fig. 5A. Higher magnification and resolution is necessary, along with statistically valid quantitative analysis of the relocation phenomenon.

- Per the suggestion, we replaced the old images with higher magnification and resolution pictures for Fig. 5A, and counted 3 – 5 fields of cells with cytoplasmic Zeb1 presence (i.e. the circular line of the nucleus has become blurred) and calculated their percentage to total number of cells (DAPI-stained) for each treatment (Fig. 5B). Based on the quantitative analysis, we have revised the text as “mRMVEC immunostaining provided evidence that Zeb1 was clearly present not only in the nucleus but also in the cytosol of about 30% cells (Fig. 5A and B). However, an addition of the ZEB1-CtBP inhibitor NSC95397 to the culture medium significantly relocated Zeb1 from the nucleus to the cytosol in almost 100% cells (Fig. 5A and B), whereas the inhibitor MTOB had little effect on the translocation of Zeb1 (Fig. 5A and B).”

In addition, the authors need to utilize an independent method to characterize the translocation- this could be to isolate nuclear and cytosolic extract and then blot for the levels of Zeb1 with relevant nuclear and cytoplasmic controls for purity of the fractionation.

- Based on the suggestion, we did an additional experiment and added following result to the text: “To validate the translocation of Zeb1 characterized by IF (Fig. 5A), we isolated both nuclear and cytosolic fractions of total protein samples from both PBS-control mRMVECs and cells treated with either MTOB or NSC95397 for Zeb1-probed WB. We also used Ctbp and Actb as relevant nuclear and cytosolic controls though Actb also showed high expression in the nucleus as reported (ref. #43) (Fig. 5C). The treatment with 10 mM MTOB had no significant effect on the relocation of Zeb1 from the nucleus to the cytoplasm whereas the treatment with 10 μ M NSC95397 however, significantly increased cytosolic Zeb1 compared to the PBS-control (Fig. 5C), which confirms the relocation of Zeb1 upon the ZEB1-CtBP inhibitor NSC95397 treatment.”

2) The authors should use IF or fractionation to test if CtBP (and specifically CtBP1 vs. CtBP2 as their localization can differ) co-relocalizes with ZEB1 after inhibitor treatment.

- The antibody we bought from Santa Cruz Biotechnology for IF and Co-IP experiments is actually against both CtBP1 and CtBP2 (Santa Cruz cat. # sc-17759). The IF images (Fig. 5A) showed that both Ctbp1 and 2 localize with Zeb1 in the nucleus and none of the Ctbp1 and 2 translocated to the cytoplasm as with Zeb1 when cells were treated with the ZEB1-CtBP inhibitor NSC95397 (Fig. 5A-B), suggesting exclusive nuclear location of Ctbp. The protein fractionation experiment also showed that Ctbp1 and 2 exclusively locate in the nucleus no matter whether nuclear Zeb1 relocates to the cytoplasm or not (Fig. 5C).

3) The authors must use genetic knockdown of CtBP1, 2, or both to substantiate their claim that CtBP inactivation relocalizes ZEB1 and alters ZEB1 level and function in CNV associated phenomena in vitro- either by si or sh techniques, including rescue of effects with rescue Ctbp1 and 2 expression, as the specificity of the compounds used has not been absolutely established in this cell system.

- We did not claim inactivation of Ctbp relocates Zeb1, what we showed was that the inhibition of Zeb1 interaction with Ctbp by the CtBP-ZEB1 inhibitor NSC95397 resulted in relocation of Zeb1 from the nucleus to the cytoplasm (Fig. 5A). In fact, the treatment with the CtBP-ZEB1 inhibitors NSC95397 and MTOB did not significantly affect Ctbp expression (Fig. 6E). It is a good suggestion for us to genetically knock down Ctbp1 and 2 to check whether or not downregulation of Ctbp1 and 2 affect Zeb1 expression and relocation, we manufactured Ctbp1 and 2 shRNA lentivirus particles released in the culture medium using a commercial mixture of Ctbp1 and 2 shRNA lentiviral vector plasmids (see Methods). We utilized Ctbp1 and 2 shRNA lentivirus to knockdown Ctbp1 and 2 in mRMVECs. About 70–80% downregulation of Ctbp1 and 2 in mRMVECs by the Ctbp1 and 2 shRNA lentivirus was validated by WB using an antibody against both Ctbp1 and 2 (Fig. 6A) as both CtBP1 and CtBP2 interact with ZEB1 and have a similar biological function as a co-repressor in the nucleus (ref. #44). Apparently, a large quantity of Ctbp was present in the nucleus compared to a relatively small amount of Zeb1 (Fig. 6A) and downregulation of Ctbp 1 and 2 did not significantly affect the Zeb1 expression detected by the WB (Fig. 6A), Zeb1 relocation (Fig. 6B and C) and cell proliferation (Fig. 6D). These results indicate that compared to Zeb1 that exists in both the nucleus and cytoplasm, Ctbp is exclusively present in the nucleus (Fig. 5A and C), and that in the nucleus Zeb1 mostly interacts with Ctbp to repress expression of genes involved in cell proliferation in mRMVECs.

4) The utility of MTOB is generally limited as it is not an “inhibitor” of CtBP- it is CtBP’s highest affinity dehydrogenase substrate (see Ref # 42)- and acts as an inhibitor at extremely high concentrations by virtue of substrate inhibition. Newer generation CtBP dehydrogenase inhibitors have been reported (for example: see Korwar et al. Bioorg Med Chem. 2016 24:2707 describing “HIPP”-class inhibitors) which may be more effective and better complement the NSC experiments in vitro and in vivo by showing efficacy of an entirely different class of CtBP inhibitor.

- We have utilized two small molecule inhibitors to reduce Zeb1 activities, we will be considering testing more inhibitors including HIPP-class molecules to regulate Zeb1 and possible applications in future experiment. Thank you for the information and suggestion!

5) Fig. 6A: The co-IP experiment lacks controls and blotting for the IP components. An IgG control IP should be shown as a negative control that the ZEB1 band seen is due to specific IP. Input lysates used in the IP's, blotting for both ZEB1 and CtBP should be shown. The CtBP blot for the IPs should be shown. Comment in the results that the CtBP antibody used recognizes both CtBP1 and CtBP2, and justify why it is not necessary to know which CtBP is binding to ZEB1 in these cells.

- Very good suggestions. We added an IgG control IP for Zeb1 (negative control), an 1/10 input lysates for Zeb1, and included a Ctpb WB using a commercial antibody (Santa Cruz cat. #: sc-17759) that recognizes both CtBP1 and CtBP2 (Fig. 6E) as both CtBP1 and CtBP2 interact with ZEB1 and have a similar biological function as a co-repressor in the nucleus (ref. #44).

6) Although the authors show evidence that NSC95397 acts by evicting CtBP from Zeb1, the molecule also inhibits other proteins, namely S100a4 (Biochemistry 2011;50:7218), which also acts as an angiogenic factor (Oncogene. 2001; 20:4685). The authors should comment in the Discussion on the potential impact of this effect of NSC95397 on an angiogenic factor vs. its effect on the the angiogenic CtBP/ZEB1 interaction.

- Thank you for referring this information. I have added following sentences in the text of Discussion: "It is of note the small molecule NSC95397 exerts multiple inhibitory function in addition to interrupt the CtBP-ZEB1 interaction, it has been reported to disrupts the S100A4/myosin-IIA interaction and inhibits S100A4-mediated depolymerization of myosin-IIA filaments. S100A4, also known as fibroblast-specific protein 1 (FSP1) is a typical marker for mesenchymal cells. Repression of ZEB1 often leads to repression of S100A4. Clearly, NSC95397 is an important small molecule that inhibits EMT through inhibition of both S100A4/myosin-IIA and CtBP-ZEB1 interaction. In addition, S100A4 was also reported to be an angiogenic factor, suggesting that NSC95397 is an anti-angiogenic substance by directly inhibiting both S100A4 and ZEB1 functions."

Minor critiques:

1) Line 121: Please cite the appropriate reference for ZEB1 knockout embryonic lethality.

- We have added the reference in the text (ref. #32). Thank you!

2) Fig. 5D. Typographical error noted in Y-axis legend.

- We corrected the figure legend as "(B) Both 10 mM MTOB and 10 μ M NSC95397 significantly reduced Zeb1 mRNA in mRMVECs and (C) cell proliferation rates, but only 10 μ M NSC95397 significantly reduced (D) cell migration, and (E) cell tube formation." Thank you!

Reviewer #2 (Remarks to the Author):

In this study, Lei Jin and colleagues demonstrated an alternative mechanism for Zeb1-regulation of corneal neovascularization. The authors showed that Zeb1 functioned as an important factor to

promote vascular EC proliferation and corneal neovascularization in a VEGF-independent manner in vitro and in vivo. Moreover, treatment with the small molecule inhibitors of CtBP, including MTOB and NSC95397, resulted in dissociation of the Zeb1-Ctbp complex and consequent upregulation of the miR-200 family, leading to inhibition in Zeb1 expression, mRMVEC proliferation and mouse corneal NV severity. However, the authors did not provide enough solid experimental evidences that can support their conclusion. The manuscript would be further improved by addressing the following issues:

1. The authors used numerous in vitro and in vivo models in their study, such as Zeb1 knockout mouse embryo lungs in Figure 1 and Zeb1-depleted MEFs in Figure 4. However, based on these results, it is hard to evaluate the role of Zeb1 in corneal ECs and neovascularization. (Zeb1 immunostaining on Zeb1-/+ cornea sections with alkali-induced CNV)

- We agree with the review on that Zeb1 immunostaining on Zeb1-/+ cornea sections with alkali-induced CNV is the best material to evaluate the role of Zeb1 in corneal ECs and neovascularization. In fact, we did double-stain sections of both Zeb1 wt (+/+) and het (-/+) corneas with and without alkali-induced CNV twice with the antibodies against Zeb1 and CD31 (endothelial cell marker). Unfortunately, we did not get Zeb1-specific staining on Zeb1-/+ corneal sections though we had a good luck with Zeb1 wt (+/+) sections (Fig. 3). It is difficult and time-consuming to re-prepare Zeb1 het (-/+) sections because breeding for Zeb1 het (-/+) mice is little bit tricky. Good news is that we successfully double-stained Zeb1 wt (+/+) sections and clearly showed that Zeb1 did co-express with Cd31 in corneal vessel cells, many more in the alkali-induced NV corneas than the PBS-control corneas (Fig. 3), indicating that Zeb1 is likely involved in CNV.

2. In Figure 1, the authors should explain how the capillary ECs and mesenchymal cells were examined and calculated in lung tissue sections just based on the H&E staining results.

- The mesenchymal cell is defined by a dark blue nucleus as indicated in Fig. 1A-C) and relatively easy to count. However, it is more difficult to define the capillary cells though we define them as the separated red areas that may contain a single or group of red blood cells and may or may not surrounded by the mesenchymal cells. Therefore, we added the following sentences in the legend of Fig. 1 to explain how the capillary ECs and mesenchymal cells were examined and calculated in lung tissue sections: “(D) more mesenchymal cells with a blue-stain nucleus (m) and less capillary cells in Zeb1^{-/-} knockout lungs. Capillary cells are defined as the separated red areas that may contain a single or group of red blood cells and may or may not surrounded by the mesenchymal cells.”

3. In Figure 3, according to IF staining, it is hard to conclude that Zeb1 expression was specifically upregulated in ECs of newly formed blood vessels in the alkali-burned mouse corneas. The results should be quantified.

- To quantitatively analyze the results as suggested by the reviewer, we re-selected 4 cryosections of two corneas of 14 days after either PBS-control or treated with NaOH, immunostained them with both Zeb1 and CD31, and counted double-stained cells in the limbus and the rest of the cornea separately. As a result, we added two more bar graphs (Fig. 3C and D) showing that the alkali treatment increased the number of Zeb1+

vascular ECs (Fig. 3C) and caused corneal NV (Fig. 3D) based on the IF images. Thank you for the suggestion!

4. In Figure 4B and F, depletion of Zeb1 resulted in significant upregulation of Vegf/Vegfr expression in MEFs and mRMVECs. Why the authors suggested that Zeb1-regulation of corneal neovascularization is independent of Vegf pathway? This should be addressed.

- Yes, in Figure 4B and F, depletion of Zeb1 resulted in significant upregulation of Vegf/Vegfr expression in MEFs and mRMVECs, suggesting that Zeb1 promotion of angiogenesis is not through up-regulation of Vegf genes because Zeb1 may actually downregulate their expression.

5. In Figure 5A, the expression of Zeb1 at the protein level and cell proliferation in mRMVECs was not significantly inhibited by treatment with MTOB or NSC95397. However, the authors showed decreased expression of Zeb1 at the mRNA level in Figure 5B and reduced cell proliferation in Figure 5C. This are confused and should be further evaluated.

- We agree with the reviewer on that MTOB or NSC95397 decreased Zeb1 mRNA (Fig. 5D), but it appeared not affect Zeb1 protein levels in mRMVECs (Fig. 5A and 6E Input). It is not surprising as the amount of a mRNA does not always agree with the amount of the related protein because protein synthesis and/or turnover may be faster or slower. Fig. 5A and C also show that NSC95397 made Zeb1 protein relocated from the nucleus to the cytoplasm and we reason that the CtBP-ZEB1 inhibitor NSC95397 disrupts the CtBP-ZEB1 complex, allowing Zeb1 “free-to-go”. Fig. 6A, the co-immunoprecipitation experiment with a commercial CtBP antibody against both CtBP1 and 2 (Santa Cruz cat. #: sc-17759) showed that Zeb1 has left the complex in the nucleus where Ctbp always presents (Fig. 5A and C), thereby the removal of Zeb1 from the nucleus leads to de-repression of Zeb1 targeted genes like Cdk inhibitors (Fig. 4F) and miR200 family (Fig. 6F) for cell proliferation (Fig. 5E).

6. In Figure 6A, the expression of Zeb1 and CtBP in the input and Co-IPs for IgG and anti-CtBP should be examined.

- Yes, we added them – thank you!

7. Considering that Zeb1 and the miR-200 family members are regulated in a negative feedback loop, it is not appropriate to simply conclude that inhibition of Zeb1 expression by MTOB and NSC95397 was mediated by the upregulation of miR-200 family members in Figure 6 and 7. This should be further addressed.

- Yes, we agree with the reviewer that more work need be done to understand how MTOB and NSC95397 down-regulate Zeb1 although through regulation of miR200 family by the inhibitors is one of possible pathways. Thank you!

Finally, we would like to take this opportunity to thank reviewers for their careful evaluation, comments, and suggestions for us to improve the manuscript. We also want to thank the editor for her/his efforts to organize the timely review and summarize reviewers’ demands for us to conduct needed experiments to satisfy for publication.

REVIEWERS' COMMENTS:

Reviewer #1 (Remarks to the Author):

The authors have addressed all concerns and critiques, and now have much better and rigorous support of their hypothesis. The revised manuscript is acceptable for publication with the only minor comment below.

The former Fig. 5D, now revised Fig. 5F still has a typo in the y-axis legend of the graph, where it says "gape" instead of "gap".

Reviewer #2 (Remarks to the Author):

The revised manuscript addresses all my concerns.

Here are the two reviewers' suggestions and my responses:

Reviewer #1 (Remarks to the Author)

The authors have addressed all concerns and critiques, and now have much better and rigorous support of their hypothesis. The revised manuscript is acceptable for publication with the only minor comment below. The former Fig. 5D, now revised Fig. 5F still has a typo in the y-axis legend of the graph, where it says "gape" instead of gap".

- I did change the typo "gape" to "gap". Thanks!

Reviewer #2 (Remarks to the Author)

The revised manuscript addresses all my concerns.

- No more revision is made. Thanks!